# Multifractal Characteristics Analysis Based on Slope Distribution Probability in the Yellow River Basin, China

**Zilong Qin [1], Jinxin Wang [2],\*  and Yan Lu [1]**

[1]  School of Water Conservancy Engineering, Zhengzhou University, Zhengzhou 450001, China;
    qinzilong@gs.zzu.edu.cn (Z.Q.); luyancn@zzu.edu.cn (Y.L.)
[2]  School of Earth Science and Technology, Zhengzhou University, Zhengzhou 450001, China
\*  Correspondence: jxwang@zzu.edu.cn; Tel.: +86-135-2687-1639

**Abstract:** Multifractal theory provides a reliable method for the scientific quantification of the geomorphological features of basins. However, most of the existing research has investigated small and medium-sized basins rather than complex and large basins. In this study, the Yellow River Basin and its sub-basins were selected as the research areas, and the generalized fractal dimension and multifractal spectrum were computed and analyzed with a multifractal technique based on the slope distribution probability. The results showed that the Yellow River Basin and its sub-basins exhibit clear multifractal characteristics, which indicates that the multifractal theory can be applied well to the analysis of large-scale basin geomorphological features. We also concluded that the region with the most uneven terrain is the Yellow River Downstream Basin with the "overhanging river", followed by the Weihe River Basin, the Yellow River Mainstream Basin, and the Fenhe River Basin. Multifractal analysis can reflect the geomorphological feature information of the basins comprehensively with the generalized fractal dimension and the multifractal spectrum. There is a strong correlation between some common topographic parameters and multifractal parameters, and the correlation coefficients between them are greater than 0.8. The results provide a scientific basis for analyzing the geomorphic characteristics of large-scale basins and for the further research of the morphogenesis of the forms.

**Keywords:** multifractal analysis; geomorphology; quantitative description; large scale; Yellow River basin

## 1. Introduction

Fractal theory was first proposed by Mandelbrot [1] to calculate the length of the British coastline. The morphological features that fill space with non-integer dimensions are called fractals. Although the morphology of landforms is complex and changeable, they are not subject to mathematical laws, forming process can be explained or described by special types of mathematical equations laws, and theorems. Fractal theory can be divided into simple fractals [2–10] and multifractals in the application of geomorphology [11–14]. The geomorphological features of the studied area can only be represented with a fractal dimension value by using simple fractal theory. For regions with similar geomorphological characteristics, the simple fractal dimension value is very close, so it is difficult to distinguish their differences [15]. Subsequently, multifractal theory was proposed to describe complex evolution processes (such as the development of river networks) and the morphological characteristics of entities in nature [16]. Multifractals are an infinite set of fractal measures comprising multiple scaling exponents. Compared to the simple fractal, the probability of topographic feature and the multifractal dimension can be calculated by using multifractal theory, which can reflect the relief of the surface of an Earth segment in a more comprehensive and detailed manner.

Since multifractal theory was first proposed, it has been widely applied in the field of geology, including soil characteristics, topographic analysis, and drainage network

extraction. Ju et al. [7] used multifractal theory to analyze the complexity of pores in different types of soil, providing an effective reference for optimizing the soil structure and improving the soil water-holding capacity. As a new method, multifractals are of great significance in the study of the relationship between soil characteristics and vegetation [13,17]. Xia et al. [18] analyzed the multifractal characteristics of soil particles under different vegetation, revealing the impact of vegetation on soil improvement, and determining the vegetation type with the best soil remediation effect. Li et al. [19] studied the relationship between soil characteristics and vegetation in mining areas by using multifractal theory, which provided a new technical support for land reclamation in mining areas. Multifractal theory also plays a very important role in geomorphological recognition. Cao et al. [11] found that multifractal technology could effectively detect active gully erosion sites. Dutta et al. [20] used multifractal techniques to distinguish between rivers and glaciers, showing that glaciers have more complex structures. Shen et al. [21] calculated the multifractal spectrum of a small basin, showing that the multifractal spectrum was more adequate to detect basin geomorphology than simple fractals. For the river network in a basin, it is the fractal structure with multifractal nature [22]. The basins of river networks also exhibit fractal characteristics, and the sub-basins of the same basin exhibit different fractal characteristics [17]. Xiang et al. [12] analyzed the spatial and temporal variations of river networks based on multifractals, showing that tributaries played a decisive role in the complex water system, and urban expansion would have a greater impact on the variation in river networks.

Previous research on watershed topography has made important advances with multifractal theory [20,23,24]. However, most of the previous research investigated the geomorphological features of small basins rather than large basins. From the perspective of geomorphology, multifractal theory can reflect the characteristics of large-scale basin landforms more comprehensively than simple fractal [25]. Therefore, it is necessary to analyze the geomorphological features of large-scale basins with multifractals, and to further verify the applicability of multifractals to basin geomorphological analysis.

In this study, the Yellow River basin was selected as the research area. This basin, which spans more than 2500 km from east to west, is the second longest river in China. The general method of this paper involved multifractal based on the slope distribution probability. Our aim was to test the feasibility of multifractal theory in large-scale basin topography, and to analyze whether the geomorphometrical features of the Yellow River Basin and its sub-basins are consistent with actual geomorphology. The scale-free range of the study area was also calculated in this paper.

## 2. Materials and Methods

### 2.1. Study Area

The study area is located between 32° N, 96° E and 42° N, 119° E, and it contains more than 370 counties in 9 provinces of China. The Yellow River rises in the Bayan Har Mountains in the Qinghai Province of China and empties into the Bohai Sea. The Yellow River is 5464 km long with a basin area of 795,000 $km^2$, and it constitutes the fifth longest river in the world and the second in China. The terrain of the study area is high in the west and low in the east. The western region mainly comprises of high mountains with an average altitude of over 4000 m a.s.l. It features perennial snow cover and developed glacial landforms. The central region is a loess landform with considerable soil erosion and an average altitude of 1000–2000 m a.s.l. The terrain of the eastern region is relatively low. The Yellow River Basin is a complex landform with a large east–west span, spanning from west to east over four landform units: the Tibetan Plateau, the Inner Mongolia Plateau, the Loess Plateau, and the Huang-Huai-Hai Plain. The Yellow River basin spans across the North China and West China continental plates; it is adjacent to the North China continental plate to the north (north of yinshan) and connected to the South China continental plate (south of Qinling-Dabie Mountains). In the upper reaches of the Yellow River basin, the continental plate of the western region formed late, and geological tectonic activity is

intense. The degree of consolidation and hardening of rock formation was low, and the fault structure was developed. The middle reaches of the study area is the main place for accumulation of loess and loess-like soil. After the formation of the Loess Plateau, the erosion was intense, and gullies developed. It is also the main sand-producing area in the Yellow River Basin. In the lower reaches, the basement structure was complex, which was buried deep, and the deformation of the cover layer was not intense. The Mesozoic red beds in the Ordos of the middle and lower reaches basin have common interlayer displacement, weak tectonic rocks, and poor anti-slip stability. In order to study the spatial heterogeneity of the geomorphology of the Yellow River Basin, it was divided into four sub-basins according to secondary basin division (this criterion was provided by the National Cryosphere Desert Data Center and was based on the catchment area of the basins, these sub-basins are the secondary basins with the largest catchment area in the Yellow River Basin): the Mainstream Basin (YRM), the Weihe River Basin (WH), the Fenhe River Basin (FH), and the Yellow River Downstream Basin (YRD) from west to east by sub-basins (Figure 1).

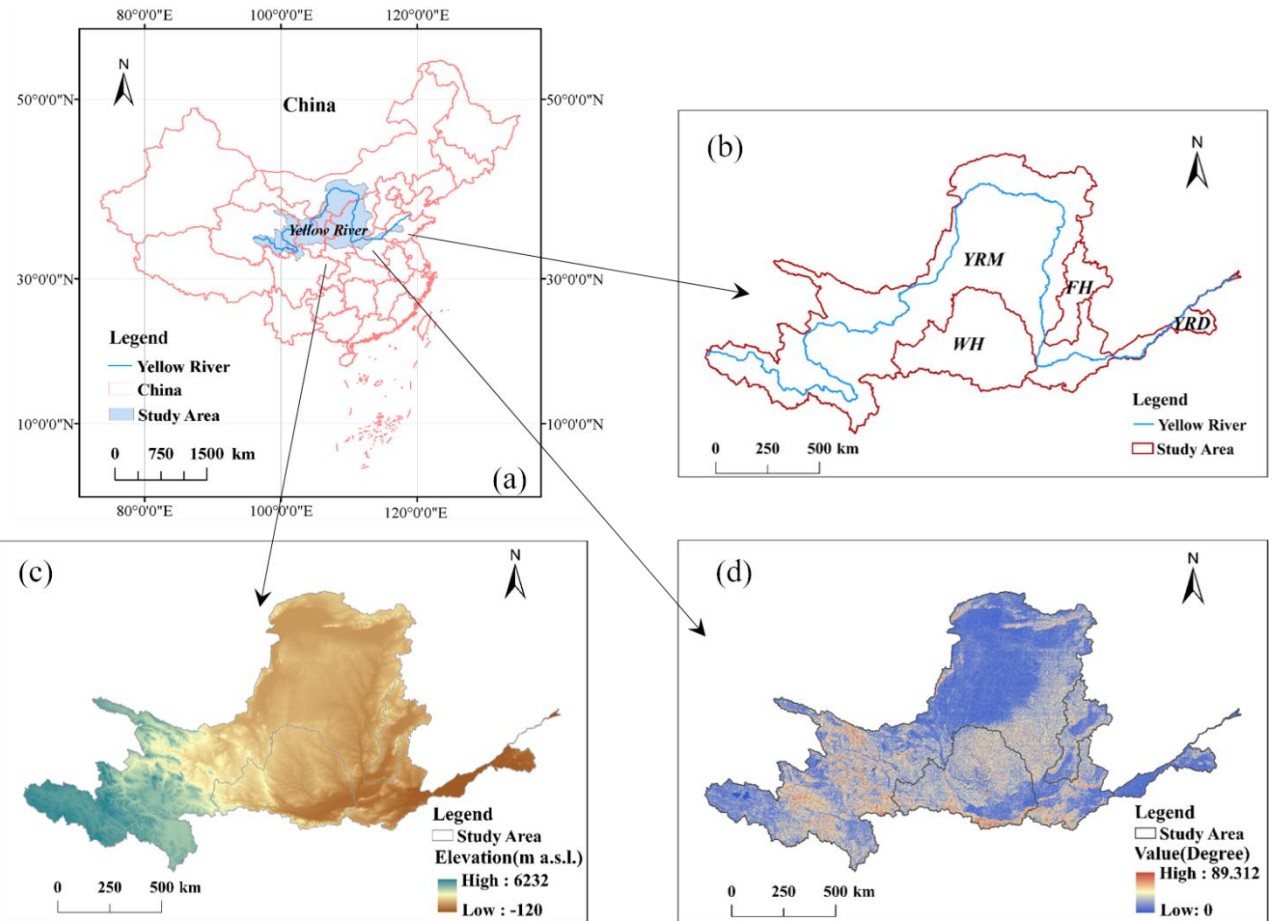

**Figure 1.** Study Area: (**a**) Location of the Yellow River Basin in China; (**b**) Location of the Yellow River sub-basins; (**c**) Digital elevation model of the Yellow River Basin and its sub-basins; (**d**) Slope of the Yellow River Basin and its sub-basins. YRM, Mainstream Basin; WH, Weihe River Basin; FH, Fenhe River Basin; YRD, Yellow River Downstream Basin.

### 2.2. Data Description

The data used in this study were extracted from a digital elevation model (DEM) with a resolution of 30 m [26,27], and the data source was ASTER GDEM. The DEM was provided by the Geospatial Data Cloud site, Computer Network Information Center, Chinese Academy of Sciences (http://www.gscloud.cn (accessed on 9 May 2021)). The data have been processed by raster mosaic and outlier correction, which is suitable for the study

of this paper. Based on the above DEM and the boundary vector data of the Yellow River Basin provided by the National Cryosphere Desert Data Center (http://www.ncdc.ac.cn (accessed on 9 May 2021), DEM of the Yellow River Basin was clipped using ArcGIS 10.4 (Figure 1c). Finally, the DEM of the Yellow River Basin was used to calculate the slope of this area (Figure 1d). The slope of the study area was calculated using Equations (1)–(3) [28]:

$$s = \arctan \sqrt{f_x^2 + f_y^2}, \tag{1}$$

$$f_x = (z_7 - z_1 + 2(z_8 - z_2) + z_9 - z_3)/(8g), \tag{2}$$

$$f_y = (z_3 - z_1 + 2(z_6 - z_4) + z_9 - z_7)/(8g), \tag{3}$$

where $s$ is the central value of the slope in the moving window, and the unit of $s$ is degree, which is a decimal; $f_x$ is the rate of elevation change in the north–south direction; $f_y$ is the rate of elevation change in the east–west direction; $z_i$ is the elevation of each point around center point 5 (Figure 2); and $g$ is the grid size of the DEM, which is shown in Figure 2.

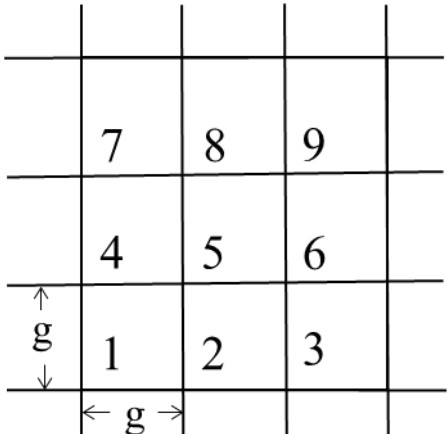

**Figure 2.** $3 \times 3$ local moving window in the digital elevation model (DEM) with a grid size of 30 m.

### 2.3. Multifractal Approach and Geomorphological Parameters

Fractal theory is widely applied to express the features of irregular and complex entities in nature. The mono-fractals can use a single fractal dimension to describe the complexity of a fractal set, which measures the amount of space filled by the fractal, disregarding local density differences. Multifractal theory accounts for the distribution of a measure ($p_i(e)$) over the fractal set and enables separating study areas with low and high intensity of the given measure [29]. Therefore, multifractals can be defined as the collection of a series of mono-fractals, each having its own singularity exponent $\alpha$ (Lipschitz–Hölder or scaling exponent) and mono-fractal dimensions [30]. The singularity exponent ($\alpha$) is determined by the probability measure ($p_i(e)$), and it can reflect the distribution of topographic features. Multifractal dimensions ($D_q$) and multifractal spectrum ($f(\alpha)$) were used to describe the features of study area, and the spectrum yields the dimensions of the fractals with the same singularity exponent.

Similar to the simple fractal, the traditional terrain parameters use a value to express the topographical features, but they cannot explain the distribution of topographical probability characteristics such as multifractal [31]. In order to analyze the relationship between the multifractal approach and the traditional geomorphological parameters, slope, topographic relief, and topographic roughness are selected for research. These geomorphological parameters can be calculated using Equations (1)–(5):

$$Z = H_{\max} - H_{\min}. \tag{4}$$

Here, $Z$ is the topographic altitude (m a.s.l.), $H_{max}$ is the maximum altitude of the study area, and $H_{min}$ is the minimum.

$$R = 1/\cos(\alpha \cdot PI/180), \tag{5}$$

$R$ is the topographic roughness [32], $\alpha$ is the average slope of the study area, and PI is the ratio of a circle's circumference to its diameter.

### 2.4. Multifractal Analysis

In this study, multifractal analysis was used to analyze the spatial distribution characteristics of the geomorphometry of the Yellow River Basin. Compared to simple fractals, multifractal analysis can express and describe the complexity of the terrain in a more detailed and continuous manner [21]. We used the generalized fractal dimension ($D_q$) and singular spectrum ($f(\alpha)$) to measure the multifractal characteristics. We adopted the fixed-size box-counting method [33] to calculate the multifractal characteristics of the study area. The study flow chart is shown in Figure 3.

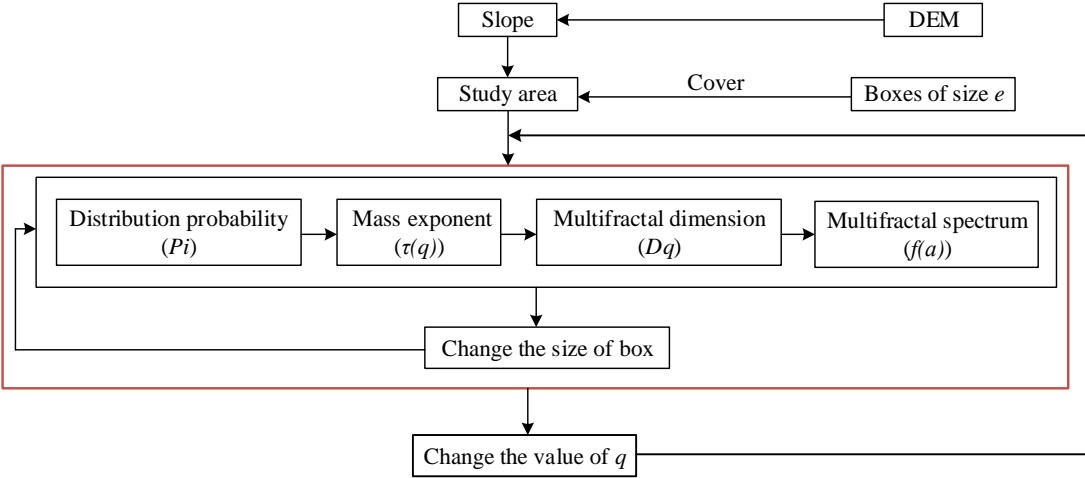

**Figure 3.** Workflow diagram of the multifractal analysis in this study.

The steps for calculating the multifractal measures of the study areas are as follows:

(1) The study area $F$ is covered with boxes of size $e \times e$, and the total number of non-empty boxes is denoted $N(e)$. $p_i(e)$ is the probability measure of the region contained in each box, i.e., the distribution probability of characteristic information. $p_i(e)$ is not the same for different units. $p_i(e)$ and $e$ are related via Equation (6):

$$p_i(e) \propto e^{\alpha}, \tag{6}$$

where $\alpha$ is a singular exponent, which corresponds to different units. $p_i(e)$ can be computed with Equation (7):

$$p_i(e) = \frac{c_i}{\sum_{i=1}^{N(e)} c_i}, \tag{7}$$

where $c_i$ is the characteristic information of terrain in box $i$ (e.g., elevation and surface area), and $\sum_{i=1}^{N(e)} c_i$ is the total characteristic information of the study area under the scale of $e$, which reflects the overall morphological features of the studied basin. In this study, $c_i$ is the sum of the slope in box $i$ and $\sum_{i=1}^{N(e)} c_i$ is the sum of the total slope of the study area. Compared to the elevation and surface area, the slope can better express the complexity of the terrain [34,35], so we took the slope as the basic data for computing multifractal parameters.

(2) The partition function $M(e, q)$ is defined as the weighted sum of the slope distribution probability $p_i(e)$ to the power of $q$ (Equation (8)):

$$M(e, q) = \sum_{i=1}^{N(e)} p_i^q(e),$$ (8)

where $q$ is the order of the statistical moment, $q \in (-\infty, +\infty)$, and is used to describe the magnitude of the heterogeneity in the multifractal analysis; a different $q$ represents the important role of different slope probability subsets in the partition function. In the calculation, we took different $q$ values and calculate the partition function $M(e, q)$ under the corresponding $q$ value. There is a good linear relationship between the logarithm of the partition function $\ln M(e, q)$ and the logarithm of box size $\ln e$ when the research object is subjected to multifractal nature.

(3) For a given moment $q$, the relationship between the mass exponential function $\tau(q)$ and $M(e, q)$ is shown in Equation (9). In the calculation, the size of the box under the corresponding $q$ value is changed, and the partition function under the corresponding box size is computed. Then, $\tau(q)$ can be computed through the coefficient of the straight line fit of $\ln M(e, q) \sim \ln e$ (Equation (10)):

$$M(e, q) \propto e^{\tau(q)},$$ (9)

$$\tau(q) = \lim_{e \to 0} \frac{\ln M(e, q)}{\ln e},$$ (10)

where $\tau(q)$ is an eigen function of the multifractal behavior. When $\tau(q)$ is a convex function with respect to $q$, the research object exhibits multifractal features.

(4) The generalized fractal dimension $D_q$ is defined as Equation (11), and $D_q$ varies with $q$. $D_{q=0}$ is the capacity dimension when $q = 0$ in multifractal analysis; $D_{q=1}$ is the information dimension, i.e., the information entropy when $q = 1$; $D_{q=2}$ is the correlation dimension when $q = 2$.

$$D_q = \begin{cases} \frac{1}{q-1} \lim_{e \to 0} \frac{\ln M(e,q)}{\ln e} = \frac{\tau(q)}{q-1} & q \neq 1 \\ \lim_{e \to 0} \frac{\sum_{i=1}^{N(e)} p_i \ln p_i}{\ln e} & q = 1 \end{cases},$$ (11)

Here, $D_q$ is usually a monotonically decreasing function with $q$. It describes the scaling behavior of the region where the probability measures are most concentrated when $q \to +\infty$ and most rarefied when $q \to -\infty$.

(5) When $\tau(q)$ is differentiable, the multifractal spectrum $f(\alpha)$ and singular exponent $\alpha(q)$ can be computed by the Legendre transformation of Equation (12).

$$\begin{cases} \alpha(q) = \frac{d\tau(q)}{dq} \\ f(\alpha) = q \cdot \alpha(q) - \tau(q) \end{cases},$$ (12)

Here, $f(\alpha)$ is usually a smooth upper convex curve. Each point on the $f(\alpha) \sim \alpha(q)$ curve represents the fractal dimension of the subset with the same singular exponent $\alpha(q)$ [23]. The $f(\alpha) \sim \alpha(q)$ curve is converted to a point in the simple fractal.

(6) In the calculation of the generalized fractal dimension and the multifractal spectrum, the value of $q$ plays a key role in the operation speed and the accuracy of the results [36–38]. Theoretically, $q \in (-\infty, +\infty)$, but in the actual calculation, only a limited range can be selected as the value of $q$. According to the research findings of [39], when the convergences coefficient $\xi < 0.2\%$, it could cause $\frac{d\alpha_{\max}}{\Delta \alpha}$ and $\frac{d\alpha_{\min}}{\Delta \alpha}$ to change very little, and the multifractal spectrum computed with $q$ in this range was considered to reflect the

multifractal characteristics of the research object. The range of $|q|$ can be determined as Equation (13).

$$\xi = \frac{|f_q - f_{q-1}|}{|f_q - f(\alpha)_{max}|} , \tag{13}$$

In this study, the range of $q$ was preset as $[-100, 100]$ and the step was $\Delta q = 1$. Then, according to Equation (13), the YRM, the WH, the FH, the YRD, and the YR were computed to determine the value of $q$. When the convergence coefficient $\xi$ met the relation of $\xi < 0.2\%$, the study area ranges of $q$ were all $[-30, 30]$; in other words, when the value of $q$ was beyond this range, the results did not satisfy the geometric features of the multifractal. Therefore, when calculating the multifractal characteristics of the study areas, the range of $q$ was $[-30, 30]$ and the step was $\Delta q = 1$.

## 3. Results

### 3.1. Determination of Multifractal Characteristics

It is necessary to test whether the studied basin possesses multifractal properties before the multifractal analysis. In this study, the logarithmic curve of the partition function $M(e, q)$ and the box size $e$ of the Yellow River Basin and its sub-basins were computed, where the box size ranged from 500 to 40,000 m with an increment of 500 m, and the value range of the order moment $q$ was $[-30, 30]$ with an increment of 1. In order to present the results in a simple and clear way, we selected only five results. The results when $q$ was $-30, -15, 0, 15,$ and $30$ were selected for exhibit in this study. As shown in Figure 4, when $q < 0$, the curve cluster has a slight fluctuation. In contrast, when $q \geq 0$, the curve is more stable and closer to the fitted line. In general, the $\ln M(e, q) \sim \ln e$ plot of the Yellow River Mainstream Basin, the Fenhe River Basin, the Weihe River Basin, the Yellow River Downstream Basin, and the Yellow River Basin has a good linear relationship, which satisfies the exponential law of Equation (9), and the Pearson correlation coefficients are all greater than 0.95. The results show that the study areas are scale invariant within selected scales; that is, the study areas exhibit clear multifractal characteristics.

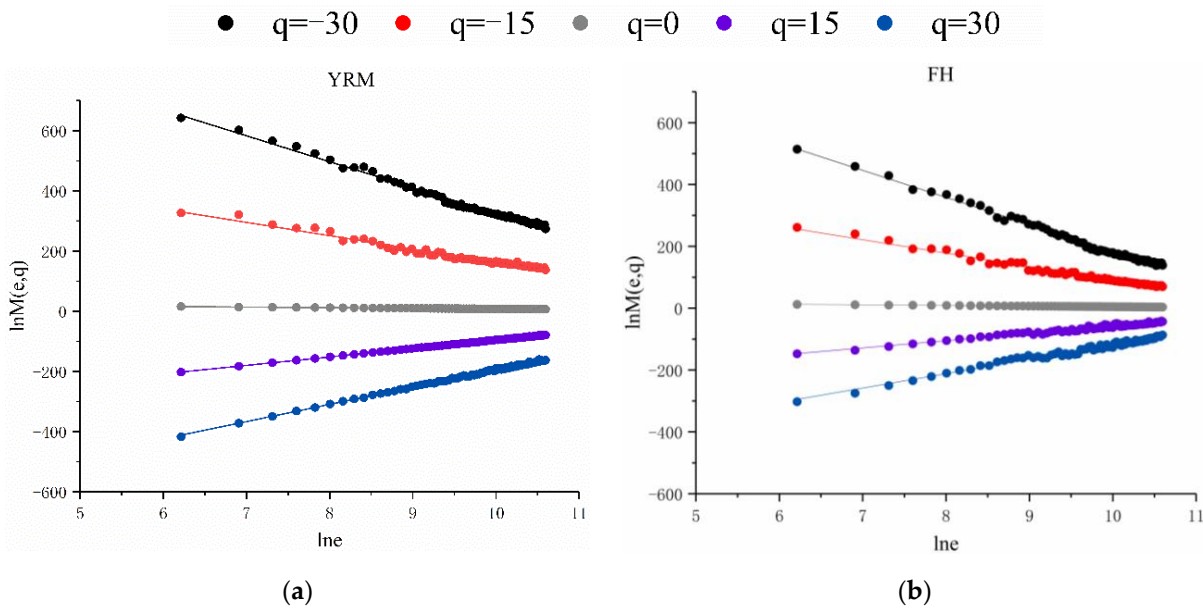

**Figure 4.** *Cont.*

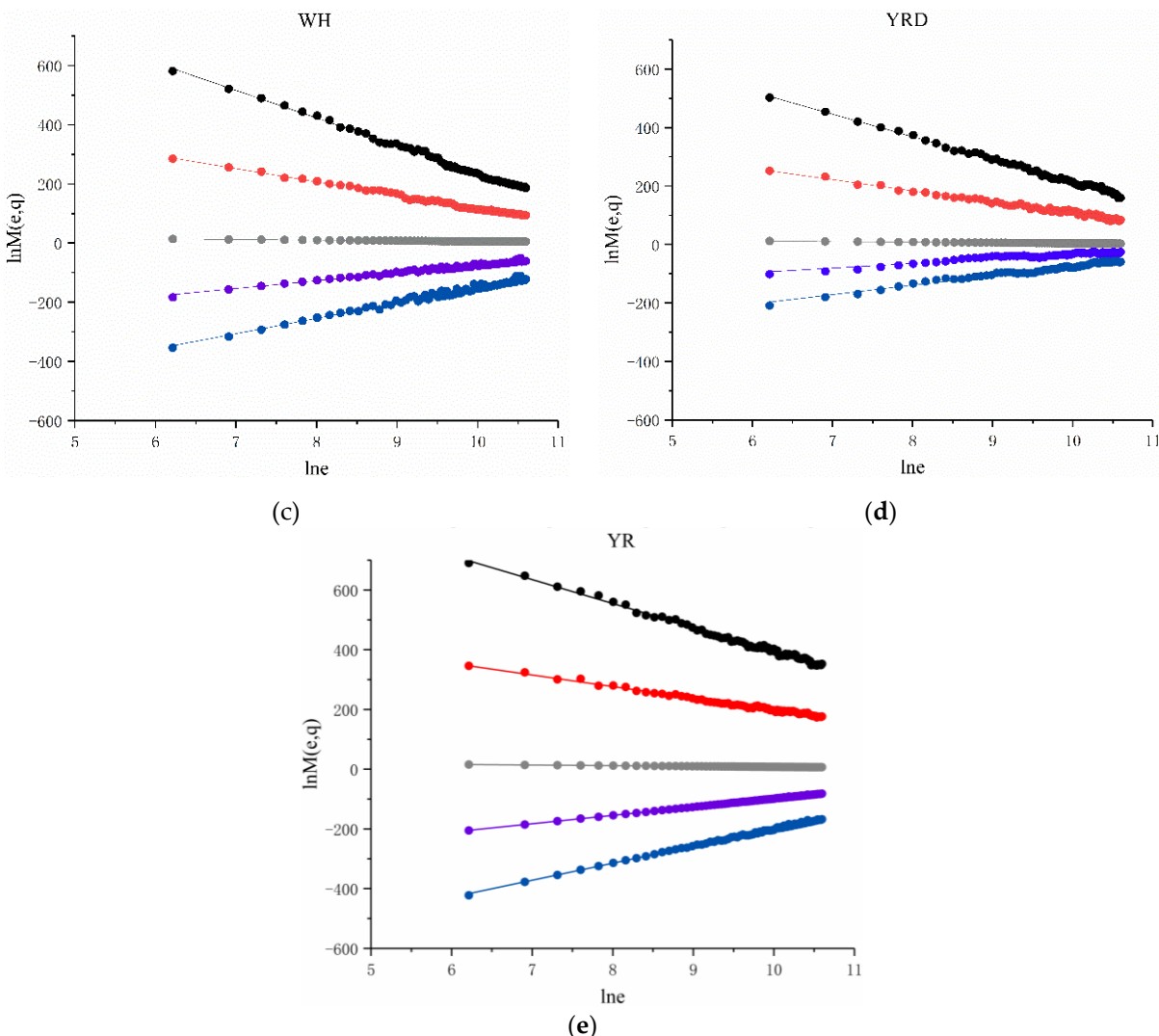

**Figure 4.** The ln $M(e, q)$~ln $e$ relationships of (**a**) The Yellow River Mainstream Basin (YRM); (**b**) The Fenhe River Basin (FH); (**c**) The Weihe River Basin (WH); (**d**) The Yellow River Downstream Basin (YRD); and (**e**) The Yellow River Basin (YR).

The relationships between the mass exponent $\tau(q)$ and moment $q$ of the main basin and sub-basins studied are shown in Figure 5. The trend lines are all upper convex curves, and $\tau(q)$ increases with $q$. When $q < 0$, the mass exponents of the sub-basins in the Yellow River Basin are YRD>FH>YRM>WH. When $q > 0$, YRM>WH>FH>YRD; when $q \in [-3, 3]$, the four curves coincide. The values of the mass exponent of the four sub-basins in the Yellow River Basin changes from west to east in space. When $q < 0$, the mass exponent gradually increases from west to east; when $q > 0$, the mass exponent gradually decreases from west to east. The trend line of mass exponent $\tau(q)$ and moment $q$ in the Yellow River Basin is similar to that in the Yellow River Mainstream Basin. The upper convex curve in Figure 5 also proves that the Yellow River Basin and its sub-basins have multifractal properties.

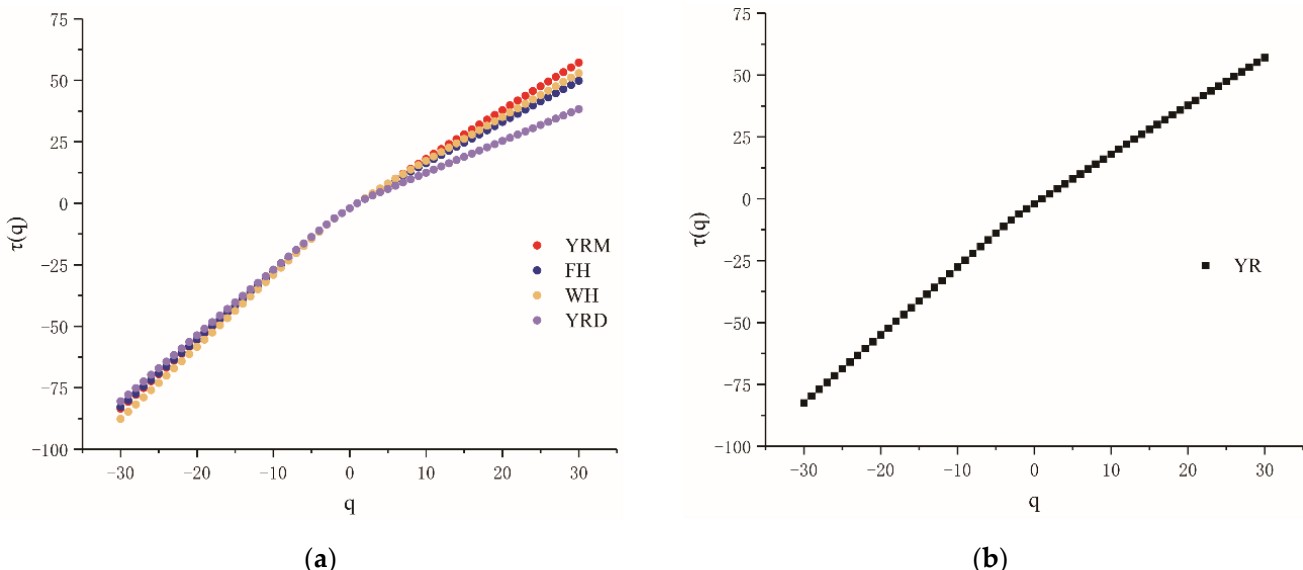

**Figure 5.** The relationships between the mass exponent $\tau(q)$ and the moment order $q$ of (**a**) The Yellow River Mainstream Basin (YRM), the Fenhe River Basin (FH), the Weihe River Basin (WH), the Yellow River Downstream Basin (YRD); and (**b**) The Yellow River Basin (YR).

### 3.2. Multifractal Dimension Analysis

The spectrum of the generalized multifractal dimension $D_q$ was computed by the least square linear regression. The generalized multifractal dimensions of the YRM, FH, WH, YRD, and YR were calculated when $q$ ranged from $-30$ to 30. The relationship of $D_q - q$ is shown in Figure 6. When $q > 1$, the generalized fractal dimension $D_q$ describes the properties of regions with higher or more concentrated probability measures. When $q < 1$, $D_q$ describes the properties of regions with lower or more sparse probability measures. Among the sub-basins of the Yellow River Basin, when $q > 1$, the value for the YRM is largest and the value for the YRD is smallest. By contrast, when $q < 1$, the value for the YRD is largest, and the value for the WH is smallest. The value of the YR is similar to that of the YRM.

As the Yellow River Downstream flows gently, the "overhanging river" (fluvial sections where the river bottom has risen above the ground level as result of silting) and deltas were formed. Its terrain is flat, open, and low-lying, dominated by small sharp fluctuations. Therefore, compared to the other regions in the YR, the value for the YRD is largest when $q < 1$ and smallest when $q > 1$. However, the FH and the WH belong to the Loess Plateau region, where the topography is relatively fragmented and the terrain is relatively high. The values for the FH and the WH are close to one another over the entire range of $q$. The YRM has a large east–west span and is dominated by a plateau topography characterized by high terrain and large fluctuations. Therefore, the value $D_q$ of YRM is largest at $q > 1$.

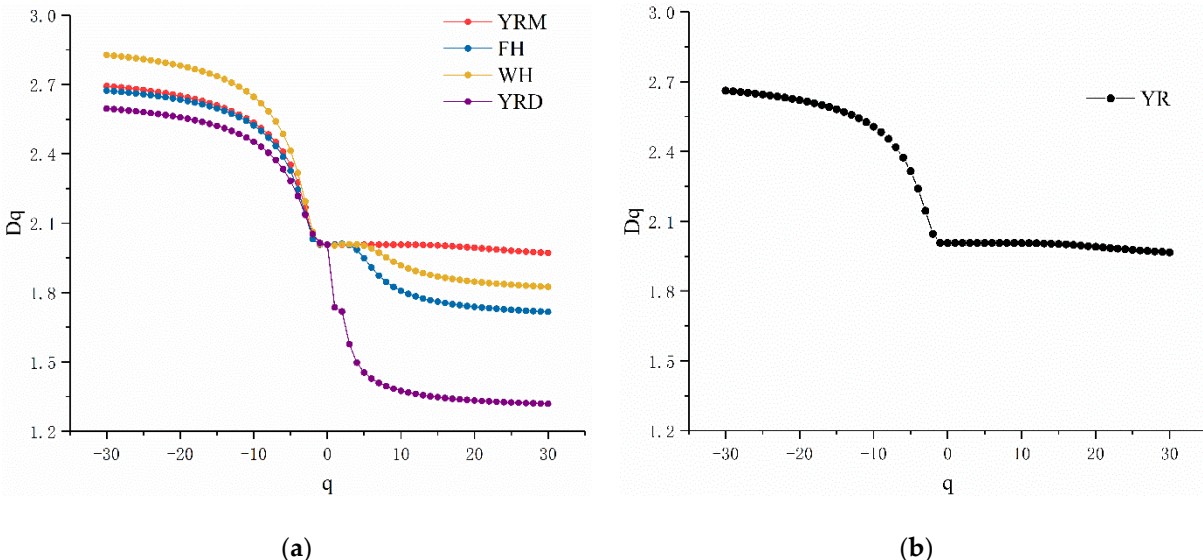

(**a**)　　　　　　　　　　　　　　　　　　　　(**b**)

**Figure 6.** The relationships between the generalized multifractal dimension $D_q$ and the moment order $q$ of (**a**) The Yellow River Mainstream Basin (YRM), the Fenhe River Basin (FH), the Weihe River Basin (WH), the Yellow River Downstream Basin (YRD), and (**b**) The Yellow River Basin (YR).

### 3.3. Multifractal Spectrum Analysis

The heterogeneity of geomorphology was mainly analyzed by the multifractal spectrum, i.e., the relationship between the multifractal spectrum $f(\alpha)$ and the singular exponent $\alpha(q)$ in Equation (12). The multifractal spectrum and its parameters of the Yellow River Basin and its four sub-basins were computed according to Equation (12), and the results are shown in Figure 7 and Table 1. A brief summary of the topographic features of the study basin and sub-basins is shown in Table 2.

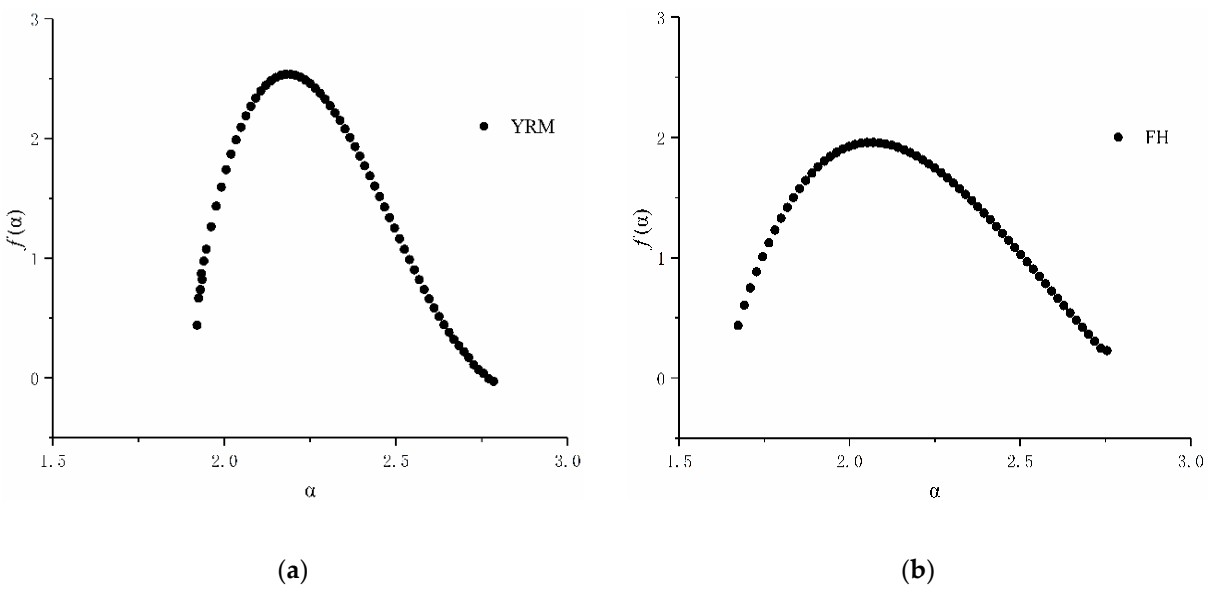

(**a**)　　　　　　　　　　　　　　　　　　　　(**b**)

**Figure 7.** *Cont.*



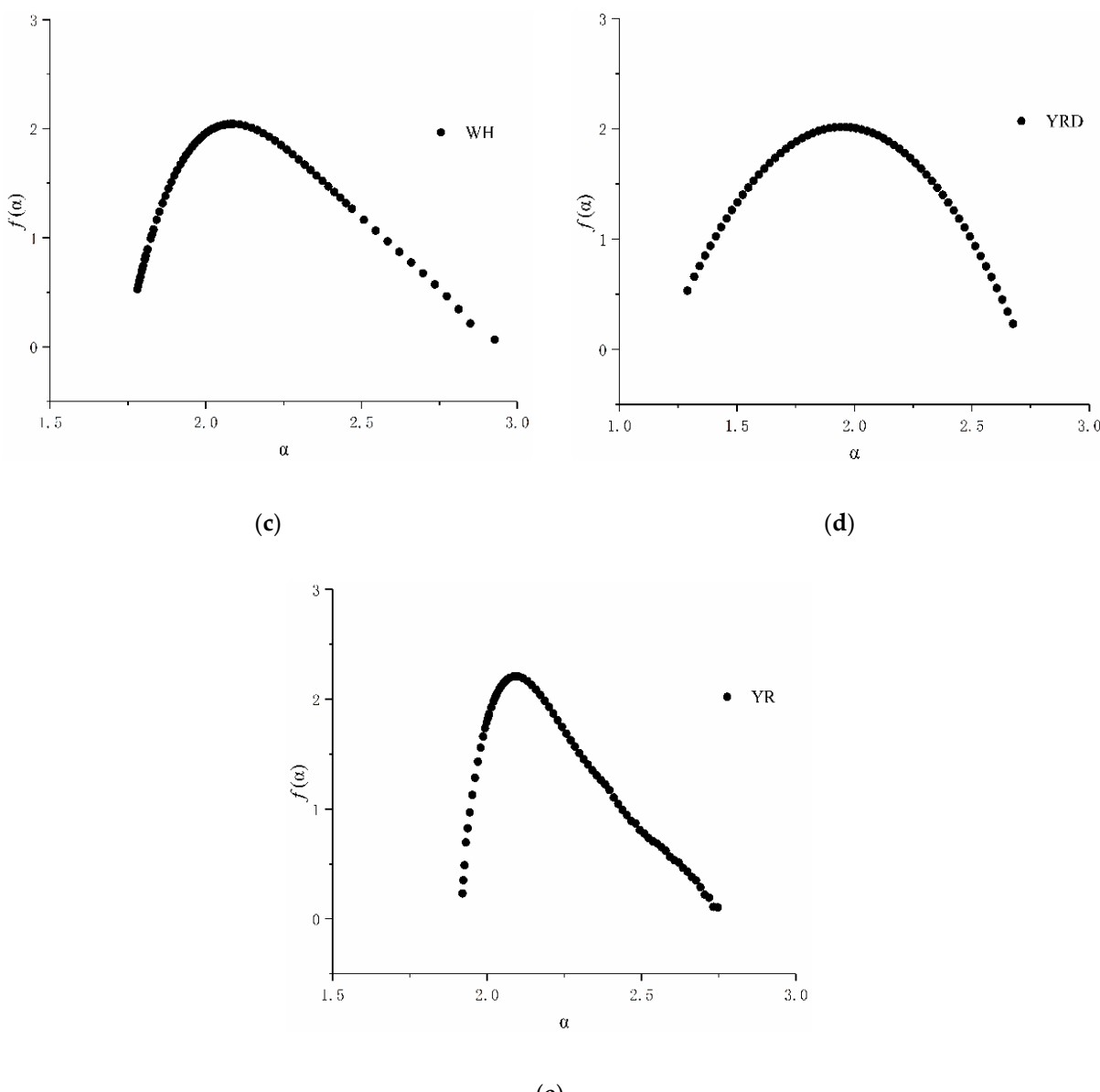

(**c**)  (**d**)

(**e**)

**Figure 7.** The multifractal spectra of (**a**) The Yellow River Mainstream Basin (YRM), (**b**) The Fenhe River Basin (FH), (**c**) The Weihe River Basin (WH), (**d**) The Yellow River Downstream Basin (YRD), and (**e**) The Yellow River Basin (YR).

**Table 1.** Multifractal characteristic parameters of the study areas.

| Study Area | $\alpha_{min}$ | $\alpha_{max}$ | $f(\alpha_{min})$ | $f(\alpha_{max})$ | $\Delta\alpha$ | $\Delta f$ |
|---|---|---|---|---|---|---|
| YRM | 1.91998 | 2.78403 | 0.44121 | 0.02846 | 0.86405 | 0.41275 |
| FH | 1.67325 | 2.75445 | 0.43686 | 0.22858 | 1.08120 | 0.20828 |
| WH | 1.78660 | 2.92725 | 0.60605 | 0.06559 | 1.14065 | 0.54046 |
| YRD | 1.28930 | 2.67504 | 0.44354 | 0.23218 | 1.38574 | 0.21136 |
| YR | 1.92088 | 2.74660 | 0.23231 | 0.10422 | 0.82572 | 0.12809 |

**Table 2.** A brief summary of topographic features of the study areas.

| Study Area | Location | $\Delta\alpha$ | $\Delta f$ | Symmetry of Multifractal Spectrum | Geomorphological Features |
|---|---|---|---|---|---|
| YRM | The northwest of the Yellow River Basin | medium | $\Delta f > 0$, large | asymmetrical | The landform of this region is relatively simple. The proportion of wave peaks is larger than valleys. The terrain is high and undulating, with high mountains as the main terrain. |
| FH | The east of the Yellow River Basin and the Loess Plateau, China | large | $\Delta f > 0$, medium | asymmetrical | The terrain is relatively complex but slightly less complex than the WH. FH has fewer large fluctuations, and most of them are low mountains and hilly areas. |
| WH | The south of the Yellow River Basin and the Loess Plateau, China | large | $\Delta f > 0$, very large | asymmetrical, and the spectrum has a longer trailing tail | The terrain is also relatively complex, and is higher, the proportion of peaks is greater than that of valleys, and the terrain is mostly dominated by fluctuations with large magnitudes. |
| YRD | The southeast of the Yellow River Basin | Very large | $\Delta f > 0$, medium | symmetrical | The distribution of the terrain is not uniform, but the relief is small. It is low-lying and hilly, and the river channel forms the overhanging river. |
| YR | Northwest China | medium | $\Delta f > 0$, medium | asymmetrical, and the spectrum has a trailing tail | The terrain is relatively complex. The proportion of peaks is greater than that of the valleys. The topography is mainly dominated by plateaus. |

Three parameters in the multifractal spectrum are important when describing the complexity (heterogeneity) of a topography: (1) The span of the singular exponent $\alpha(q)$ is the width of the multifractal spectrum $\Delta\alpha$. $\alpha(q)$ indicates the level of relief inhomogeneity, irregularity, and complexity in each sub-region within the basin, which is also the difference distribution of the Earth's surface or landforms. $\alpha_{\min}$ and $\alpha_{\max}$, respectively, indicate the singular exponent of the distribution probability of maximum characteristic information $p_i(e)_{\max}$ and the distribution probability of minimum characteristic information $p_i(e)_{\min}$ with the change in $e$. The smaller the $\alpha_{\min}$, the larger the $p_i(e)_{\max}$; conversely, the larger the $\alpha_{\max}$, and the smaller the $p_i(e)_{\min}$. Therefore, we can use the span of the singular exponent $\Delta\alpha = \alpha_{\max} - \alpha_{\min}$ to describe the unevenness in the distribution probability of the basin characteristic information. A larger $\Delta\alpha$ indicates that the distribution of characteristic information in the basin is less uniform, the fluctuations in the geomorphic surface are greater, the internal difference in the research object is greater, and the polarization trend of each subset probability is clearer. On the contrary, a smaller $\Delta\alpha$ indicates that the geomorphometric difference is smaller inside the fractal body, the topography is simpler, and the distribution of subsets tends to be concentrated and uniform. (2) The difference $\Delta f$ between the maximum and minimum values of the multifractal spectrum is $f(\alpha)$. $f(\alpha_{\min})$ and $f(\alpha_{\max})$, respectively, represent the number of subsets of the maximum and minimum probabilistic characteristic information. The difference of $\Delta f = f(\alpha_{\min}) - f(\alpha_{\max})$ can be used to calculate the difference between the maximum and minimum distribution probability subset numbers of the basin characteristic information, which indirectly reflects

the proportion of the number of peaks and valleys on the basin geomorphology. When $\Delta f < 0$, the $f(\alpha) - \alpha(q)$ curve is hooked to the right, and the number of grid points contained in the maximum characteristic information distribution probability subset is less than the minimum probability subset number. The gully area of the basin is large, and the geomorphological morphology is sharper. On the contrary, when $\Delta f > 0$, the curve is hooked to the left, and the geomorphological shape is more rounded. When $\Delta f = 0$, the curve $f(\alpha) - \alpha(q)$ is symmetrical and bell-shaped. (3) Symmetry of curve $f(\alpha) - \alpha(q)$. The multifractal spectrum is more symmetrical, which indicates that the distribution proportion of each landform type is more uniform in the study areas.

Figure 7 and Table 1 show that the singular exponential span of the YRD $\Delta\alpha$ = 1.38574 is largest, that is, the multifractal spectrum has a large distribution range, and the $\Delta f$ of 0.21136 is small. The above results indicate that the distribution of the terrain in this region is not uniform, but the relief is small. The YRD is located in the southeast of the Yellow River Basin, and it is low-lying and hilly. Due to the large amount of silt in the Yellow River water, it quickly deposited in the lower reaches, and the river channel is continuously silted up, forming the "overhanging river". Compared to the other regions, the area of the YRD is smallest, and there are only two types of landforms: the flat beach and the low hills. There are no steep mountains, and there is no transitional terrain except those two landforms, which lead to the result that the distribution of landforms is relatively concentrated and the "polarization" is clear. The symmetry of the multifractal spectrum in the YRD is relatively uniform (Figure 7d), which also indicates that there are no very steep mountains and the distribution of area of the two landforms is close. The singular exponent span $\Delta\alpha$ of the WH is second only to that of the YRD, which is 1.14065, indicating that the terrain in this region is relatively complex. Unlike the YRD, the right side of the multifractal spectrum curve of the WH has a longer trailing tail, which is of poor symmetry (Figure 7c), indicating that the terrain in this region is higher, the proportion of peaks is greater than that of valleys, and the terrain is mostly dominated by fluctuations with large magnitudes. $\Delta f$ of the WH is largest among the study areas, which also explains the above phenomenon. The Weihe River is the largest tributary of the Yellow River. The WH is located in the south of the Yellow River Basin and the Loess Plateau, China, mainly in the gully region of the Loess Plateau.

The singular exponent span $\Delta\alpha$ of the FH is 1.0812, which is similar to the WH value and is relatively large, indicating that the terrain in this region is also relatively complex, but slightly less complex than that of the WH. The value $\Delta f$ of the FH is smallest among the several sub-basins of the YR, indicating that although the terrain of this region is complex, compared to the other basins, it has fewer large fluctuations, and most of them are low mountains and hilly areas. The symmetry of its multifractal spectrum curve is better than that of the WH (Figure 7b), which also indicates that its topography has fewer gullies than that of the WH. The Fenhe River is the second largest tributary of the Yellow River with significant soil erosion, which is located in the east of the Yellow River Basin and the Loess Plateau, China; the terrain is mainly loess hilly and mountainous regions. The value $\Delta\alpha$ of the YRM is 0.86405, which is smallest among the sub-basins, indicating that compared to the other basins, the topography and landform of this region are relatively simple. The value of $\Delta f$ is second only to that of the WH, indicating that the proportion of wave peaks in this region is larger than that of valleys. The terrain is high and undulating, with high mountains as the main terrain. The YR has a large east–west span, and it mainly passes through the plateau region. The value of $\Delta\alpha$ is 0.82572, indicating that the terrain in the YR is relatively complex, and $\Delta f$ is 0.12809, which is greater than 0. In general, the proportion of peaks is greater than that of the valleys in this region. The multifractal spectrum exhibits asymmetry and clear trailing on the right side (Figure 7e). The above phenomenon is due to the YR passing through four geomorphic units: the Tibetan Plateau, the Inner Mongolia Plateau, the Loess Plateau, and the Huang-Huai-Hai Plain. The topography is mainly dominated by plateaus and fluctuations with large magnitudes.

## 4. Discussion

In this paper, we studied some aspects of the geomorphometry of the Yellow River Basin and its sub-basins based on multifractal theory, and we analyzed its multifractal features. We also compared the results with the actual topography of the study areas, and consistent conclusions were obtained. In the multifractal analysis of the basin relief, the selection of the analysis scale is important. When the analysis scale is sufficiently large, the research objects will no longer show fractal characteristics. Only when the analysis scale is smaller than a certain value will the research objects show fractal characteristics. Therefore, it is necessary to determine the scale-free interval and to compute the fractal dimension of the study areas. The scale-free interval can also be considered as the interval in which the double logarithm conforms to the linear relation, beyond which the fractal of the research objects has no meaning. In the analysis of topographic and geomorphic features, we can study it more comprehensively by combining it with the common topographic parameters.

### 4.1. Analysis Scale of Multifractal

The selection of the analysis scale is important in multifractal analysis. If the analysis scale is too small, the sample data will not be sufficient to calculate multifractal characteristics, whereas if the analysis scale is too large, the research object will not exhibit fractal characteristics [11]. In this study, the experiment was carried out with 500 m as the starting value and 500 m as the increment. It was found through experimental calculation that when the analysis scale was larger than 40,000 m, the research object no longer exhibited multifractal properties; that is, the calculated multifractal parameters began to appear abnormal. Taking the Yellow River Basin as an example, the relationship between $\ln M(e, q)$ and $\ln e$ is shown in Figure 8. When the analysis scale $e$ was greater than 40,000, the value of the partition function appeared abnormal, and there was no clear linear relationship between $\ln M(e, q)$ and $\ln e$ (Figure 8b,c). The results for other areas were shown in Figures A1–A4 in Appendix A.

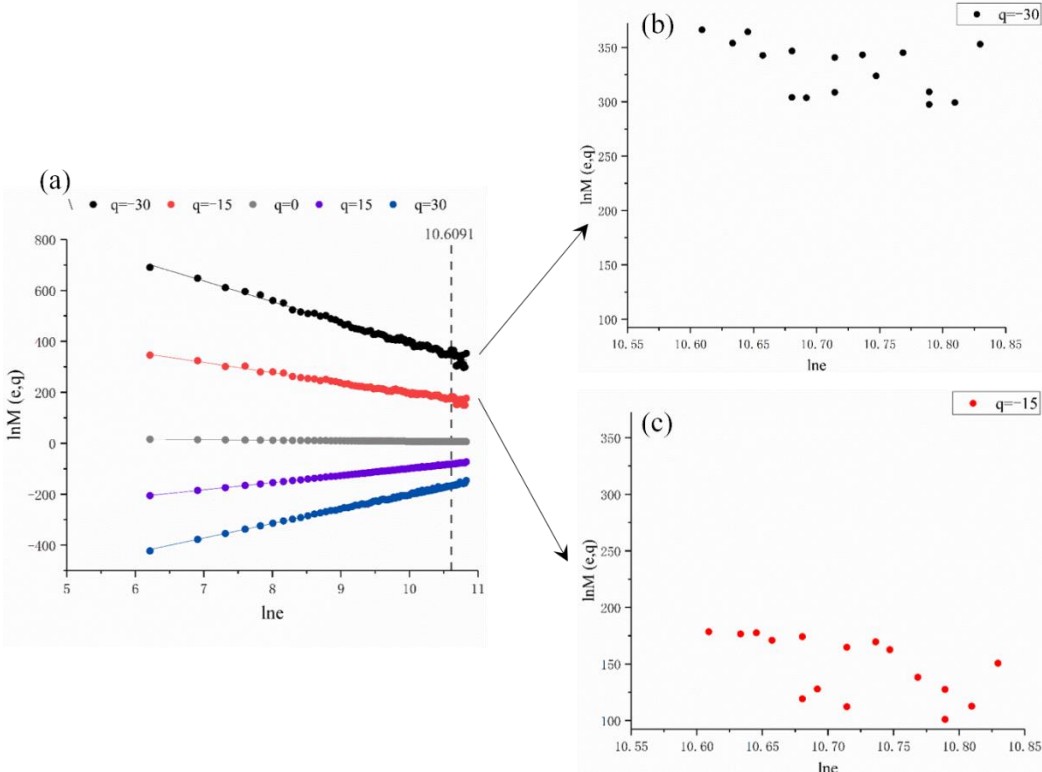

**Figure 8.** The relationship between $\ln M(e, q)$ and $\ln e$ in the Yellow River Basin. The relationship between $\ln M(e, q)$ and $\ln e$ (**a**) When $e$ is [500, 50,000]; (**b**) When $e$ is [40,000, 50,000] and $q$ is $-30$; (**c**) When $e$ is [40,000, 50,000] and $q$ is $-15$.

Therefore, the analysis scale range in this study was determined as [500, 40,000] with an increment of 500 m. Within this scale range, the partition function $M(e, q)$ has a good linear relationship with the logarithmic curve of the analysis scale $e$, and $\tau(q)$ is a convex function with $q$, that is, the selected scale range is a scale-free interval. The study areas exhibited multifractal properties within this range.

### 4.2. Common Topographic Parameters and Extra Multifractal Parameters

Geomorphometry is the science of topographic quantification, and its operational focus is the extraction of land surface parameters and objects from digital elevation models [39]. In the field of topographic analysis, most of the existing research used a terrain index or parameter to quantitatively study its characteristics. Common topographic parameters [27,28] mainly include slope, slope aspect [40,41], topographic height, topographic roughness [42,43], and fractal dimension [44]. These parameters can be extracted to analyze the features of landforms to further reveal the process and trend of geomorphology evolution.

In order to analyze the topographic features and the differences between the multifractal analysis and common topographic parameters in the topographic analysis, some common topographic indexes and other multifractal parameters were selected for research, which included topographic altitude, topographic roughness, simple fractal dimension, the multifractal parameters of the difference ($D_{q=0} - D_{q=1}$) in capacity dimension $D_{q=0}$ and information dimension $D_{q=1}$, and the ratio $D_{q=1}/D_{q=0}$. In the above parameters, slope is the average slope within the study area (Equations (1)–(3)), topographic altitude is the difference between the maximum and minimum values of elevation within the study area (Equation (4)), and topographic roughness is the reciprocal of the slope cosine of the study area (Equation (5)). In the multifractal calculation, $f(\alpha)_{\max}$ is equal to the simple dimension $D$ of the basin geomorphometric characteristics, which is the overall and comprehensive approximate representation of the basin's geomorphic morphology. Therefore, the simple fractal dimension is represented by $f(\alpha)_{\max}$ in the multifractal spectrum. Smaller $D_{q=0} - D_{q=1}$ and $D_{q=1}/D_{q=0}$ values close to 1 indicate a more uniform terrain [7]. The results are presented in Table 3.

**Table 3.** Common topographic parameters and extra multifractal parameters.

| Study Areas | Slope (°) (s) | Topographic Altitude (m a.s.l.) (Z) | Topographic Roughness (R) | Fractal Dimension (D) | $D_{q=0}$ | $D_{q=1}$ | $D_{q=0} - D_{q=1}$ | $D_{q=1}/D_{q=0}$ |
|---|---|---|---|---|---|---|---|---|
| YRM | 10.3260 | 6295 | 1.0164 | 2.5354 | 2.0086 | 2.0082 | 0.0004 | 0.9998 |
| FH | 11.2042 | 2807 | 1.0194 | 2.0058 | 2.0065 | 2.0063 | 0.0002 | 0.9999 |
| WH | 13.3426 | 3936 | 1.0277 | 2.0107 | 2.0071 | 2.0030 | 0.0041 | 0.9979 |
| YRD | 5.4959 | 1656 | 1.0046 | 2.0016 | 2.0063 | 1.7359 | 0.2704 | 0.8652 |
| YR | 10.0168 | 6352 | 1.0154 | 2.3294 | 2.0068 | 2.0069 | −0.0001 | 1.0000 |

Table 3 shows that the order of the topographic altitude and simple fractal dimension value for the sub-basins is as follows: YRM>WH>FH>YRD, which indicates that the simple fractal dimension correlates positively with the altitude (m a.s.l.), and this result is consistent with the research by Zhu [45]. The order of the slope and topographic roughness are as follows: WH>FH>YRM>YRD. The degree of $D_{q=1}/D_{q=0}$ close to 1 is: FH>YRM>WH>YRD, and the order of $D_{q=0} - D_{q=1}$ from large to small is the same as that of $D_{q=1}/D_{q=0}$, which indicates that the topography of the Fenhe River Basin is relatively uniform, while the topography of the Yellow River Downstream Basin is dispersed with poor uniformity.

In order to further study the relationship between common topographic parameters (s, Z, and R) and fractal dimensions (D, $D_{q=0}$, and $D_{q=1}$), the three-level sub-basins (a total of 31) of the Yellow River Basin were selected for computing and analyzing. We analyze

the relationship between common topographic parameters and fractal dimensions, and the results are shown in Figure 9 and Table 4.

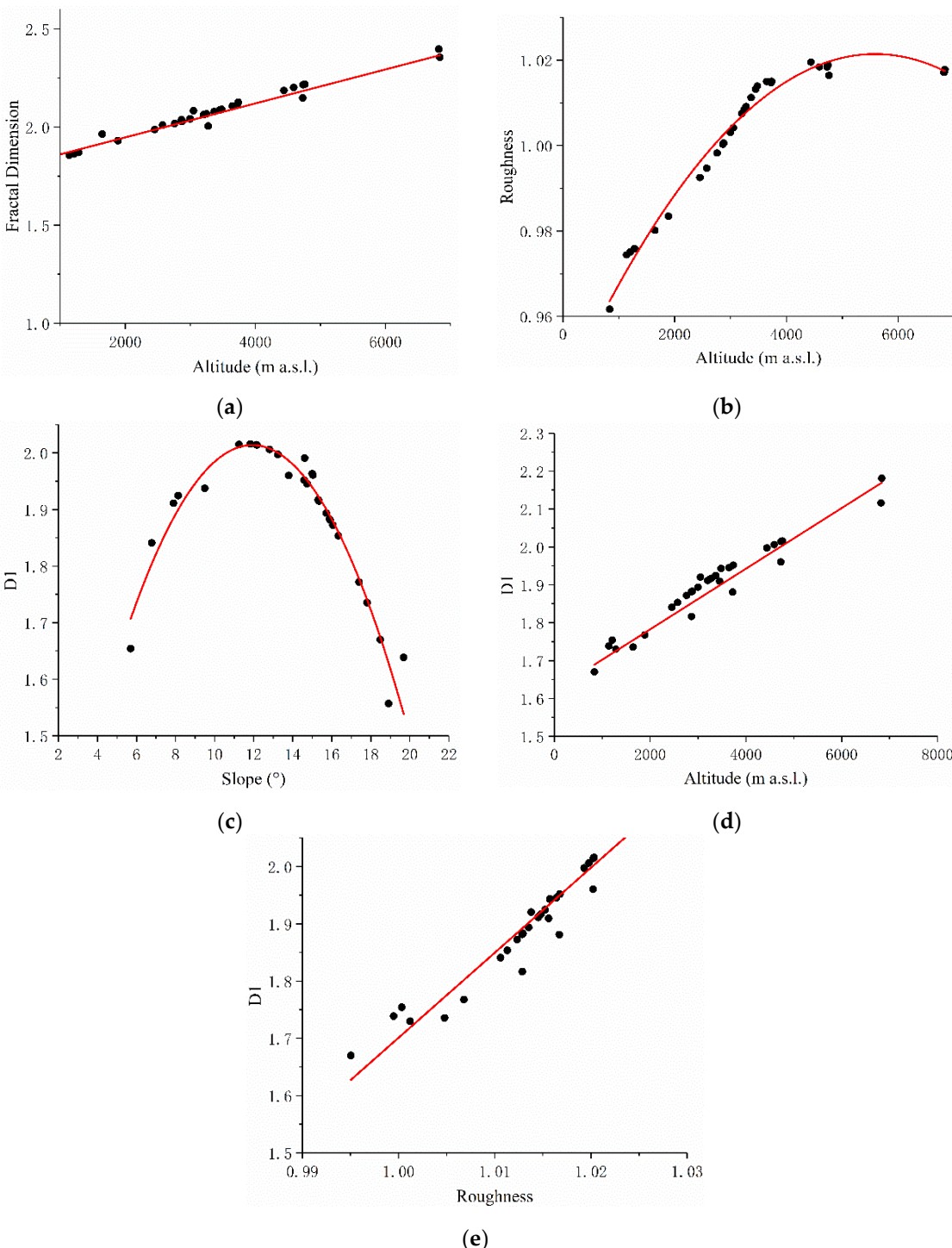

**Figure 9.** The correlation between (**a**) Altitude (m a.s.l.) and simple fractal dimension; (**b**) Altitude and topographic roughness; (**c**) Slope and information dimension; (**d**) Altitude and information dimension; and (**e**) Topographic roughness and information dimension.

**Table 4.** Pearson's correlation coefficients between general terrain parameters and other multifractal parameters.

| | Slope | Topographic Altitude | Topographic Roughness | Fractal Dimension | $D_{q=0}$ | $D_{q=1}$ |
|---|---|---|---|---|---|---|
| Slope | 1.000 | 0.392 | 0.989 ** | 0.142 | 0.314 | 0.893 * |
| Topographic Altitude | 0.392 | 1.000 | 0.979 ** | 0.876 * | 0.698 | 0.812 |
| Topographic Roughness | 0.989 ** | 0.979 ** | 1.000 | −0.072 | 0.237 | 0.919 * |
| Fractal dimension | 0.142 | 0.876 * | −0.072 | 1.000 | 0.862 | 0.327 |
| $D_{q=0}$ | 0.314 | 0.698 | 0.237 | 0.862 | 1.000 | 0.384 |
| $D_{q=1}$ | 0.893 * | 0.812 | 0.919 * | 0.327 | 0.384 | 1.000 |

** The correlation was significant at the 0.01 level (double-tailed). * The correlation was significant at the 0.05 level (double-tailed).

As can be seen from Figure 9 and Table 4, the correlations between topographic altitude and simple fractal dimension, altitude, and topographic roughness are high, which indicate that they have good linear or quadratic functional relationships (Figure 9a,b). The correlations between altitude and slope, simple fractal dimension and slope, and simple fractal dimension and topographic roughness are poor, which indicates that different topographic parameters reflect different geomorphic characteristic information. Slope, topographic altitude, and roughness all have a high correlation with information dimension $D_{q=1}$ (Figure 9c–e). In the multifractal calculation, the information dimension ($D_{q=1}$) was calculated when $q = 1$, and it is a fractal dimension to describe the characteristics of objects and can represent the degree of inhomogeneity. This shows that multifractal analysis can not only reflect the morphological characteristics of landforms but also contain the information expressed by traditional topographic parameters.

The above results show that when a single topographic parameter is used to express topographic features, it is similar to the simple fractal dimension ($D$), in that only one number is used to represent the features of a region broadly, and the information expressed is not comprehensive and specific. Some topographic parameters are highly correlated with multifractal parameters, indicating that fractal dimensions can be used to replace some topographic parameters in some way. Continuous multifractal spectrum and generalized fractal dimension are used to describe terrain features with multifractal theory. In addition to basic terrain features, it can also express the probability distribution of topographic characteristics. Multifractal can reflect topographical features more comprehensively by calculating mono-fractals at different scales, which is more superior to the traditional topographic parameters in the quantitative expression of topographic features. This paper also proves this conclusion by comparing the results of multifractal analysis with the actual topography.

## 5. Conclusions

In this study, (1) the slope database was extracted from digital elevation model (DEM) with a resolution of 30 m, and some common topographic parameters, i.e., the generalized fractal dimension ($D_q$) and multifractal spectrum ($f(\alpha)$), were computed. (2) We analyzed the multifractal characteristics of topography located in the Yellow River Basin and its sub-basins, China. The specific conclusions are as follows:

(1)    The topography of the Yellow River Basin and its sub-basins exhibited significant multifractal characteristics. The multifractal spectrum showed a hook curve to the left, which indicates that the proportion of the steep slope was large. The relief heterogeneity of the sub-basins was in descending order: the Yellow River Downstream Basin, the Weihe River Basin, the Yellow River Mainstream Basin, and the Fenhe River Basin. This shows that multifractal theory can be applied well to the research of large-scale basin geomorphology landforms or surfaces, and the feasibility of this conclusion has been verified well.

(2)   The geomorphometric features of the study areas were obtained using the multifractal theory based on slope distribution probability model, and the results were consistent with the actual terrain. In particular, the Yellow River Downstream Basin had a unique "overhanging river" morphology, leading to strong non-uniformity of the terrain in this region, and this result is consistent with the conclusion obtained by using the multifractal analysis based on slope distribution probability in this study.

(3)   The scale-free range of the Yellow River Basin was [500, 40,000] (m) with experiments based on the multifractal theory.

(4)   We analyzed the relief, surface, and landforms of the Yellow River Basin and its sub-basins by combining the common topographic parameters with other multifractal parameters. It was found that multifractal analysis can be used to describe the characteristics of the topography, and there is a strong correlation between common topographic parameters and multifractal parameters.

In this paper, multifractal analysis was applied to analyze the geomorphology of a large-scale basin and its sub-basins (small and medium-sized basins), and the results were consistent with the actual geomorphology. Therefore, the method proposed is applicable to the geomorphological basins of any scale. Since data availability is limited, DEM with resolution of 30 m was used to perform calculations; however, the study would have benefitted from the use of data with higher resolution. In addition, the study area of this paper includes topographic units of plateau, plain, hill, and basin, and the applicability of the method for other topographic types has not been tested yet. The results of multifractal analysis are meaningful only when they are calculated in the free-scale range, being therefore necessary its previous calculation. In this study, we only used the fixed-size box-counting algorithm to calculate the multifractal spectrum and dimension, and we did not use other methods, including the barycentric fixed-mass method [46] and the sandbox method, to make a comparative analysis. We will consider adopting the fixed-mass method and the sandbox method for further research in future work. We will also study the geomorphological classification, geomorphological identification, and other aspects based on the multifractal theory.

In general, the relief of the studied basin is complex, and multifractal analysis provides an effective method to quantitatively express their characteristics. The surface development is determined by many factors, such as climate, rivers, and soil. Therefore, in the further study, the internal and external factors driving geomorphic development will be explored by considering multifractal theory in the Yellow River Basin.

**Author Contributions:** Conceptualization, Zilong Qin and Jinxin Wang; methodology, Zilong Qin; software, Zilong Qin; resources, Jinxin Wang; writing—original draft preparation, Zilong Qin; writing—review and editing, Zilong Qin, Jinxin Wang and Yan Lu. All authors have read and agreed to the published version of the manuscript.

**Funding:** This research was funded by the 2018 Financial Planning Project of Henan Provincial Bureau of Geology and Mineral Exploration and Development of China, grant number HNGM2018103.

**Institutional Review Board Statement:** Ethical review and approval were waived for this study, due to the data being sourced from an open platform. Any mapper contributing to the dataset accepts its licence that clearly states that any use of the data is allowed.

**Data Availability Statement:** Publicly available datasets were analyzed in this study. This data can be found here: (1) DEM was provided by Geospatial Data Cloud site, Computer Network Information Center, Chinese Academy of Sciences: http://www.gscloud.cn (accessed on 9 May 2021). (2) The boundary vector data of the Yellow River Basin was provided by National Cryosphere Desert Data Center: http://www.ncdc.ac.cn (accessed on 9 May 2021).

**Acknowledgments:** We would like to thank all editors and reviewers for their profound comments, which help us to improve the quality of this paper.

**Conflicts of Interest:** The authors declare no conflict of interest.

**Appendix A**

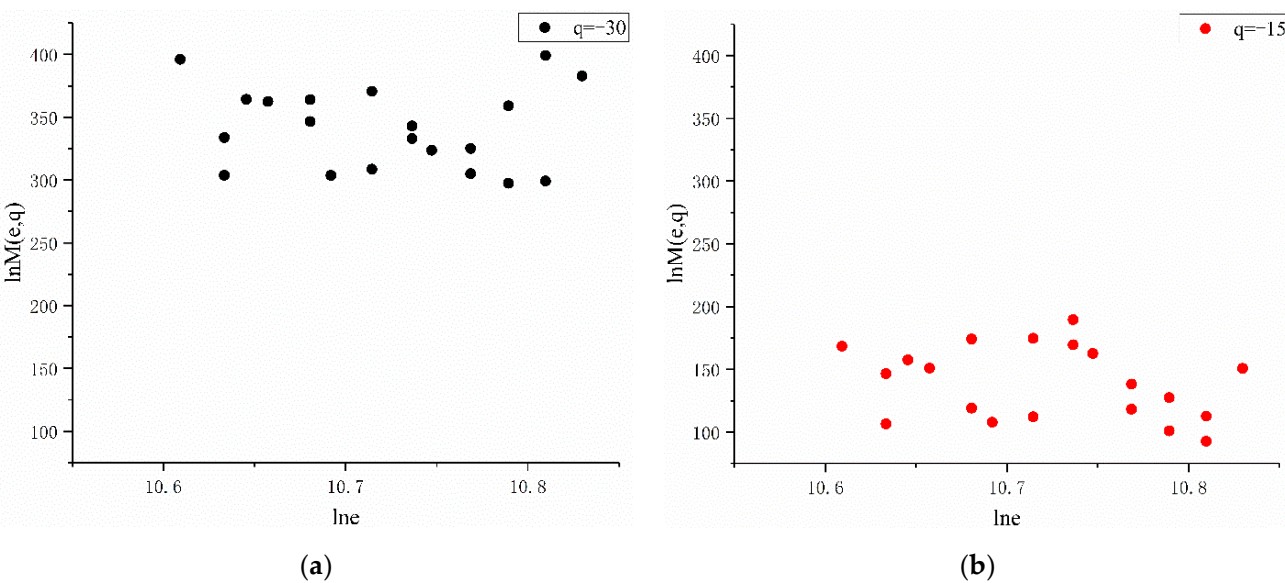

**Figure A1.** The relationship of $\ln M(e,q)$ and $\ln e$ in the Yellow River Mainstream Basin. (**a**) The relationship of $\ln M(e,q)$ and $\ln e$ when $e$ is [40,000, 50,000] and $q$ is $-30$; (**b**) The relationship of $\ln M(e,q)$ and $\ln e$ when $e$ is [40,000, 50,000] and $q$ is $-15$.

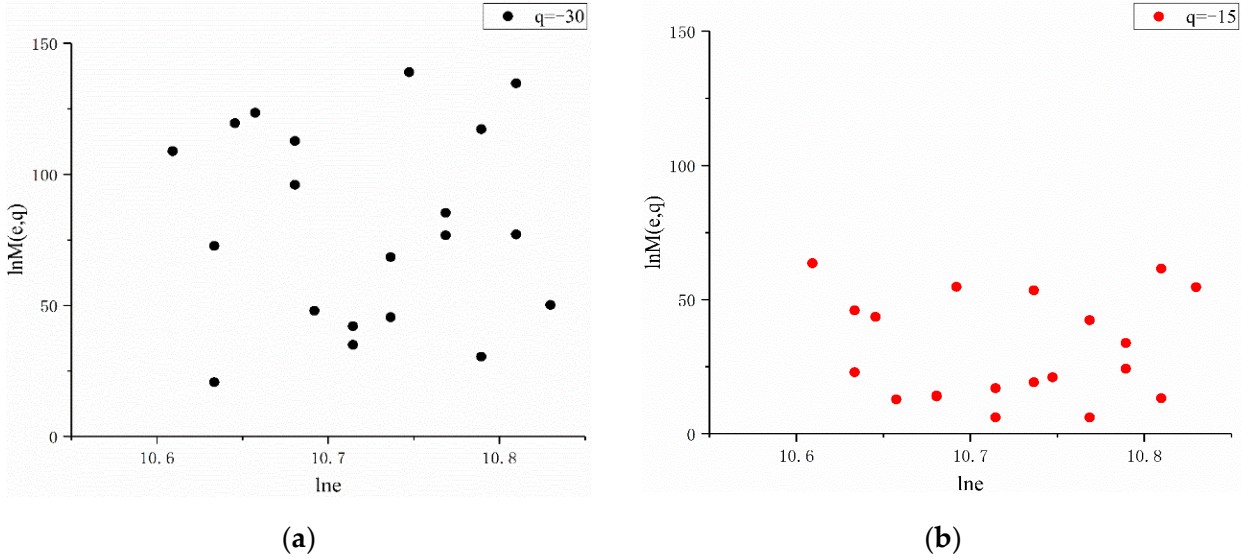

**Figure A2.** The relationship of $\ln M(e,q)$ and $\ln e$ in the Fenhe River Basin. (**a**) The relationship of $\ln M(e,q)$ and $\ln e$ when $e$ is [40,000, 50,000] and $q$ is $-30$; (**b**) The relationship of $\ln M(e,q)$ and $\ln e$ when $e$ is [40,000, 50,000] and $q$ is $-15$.

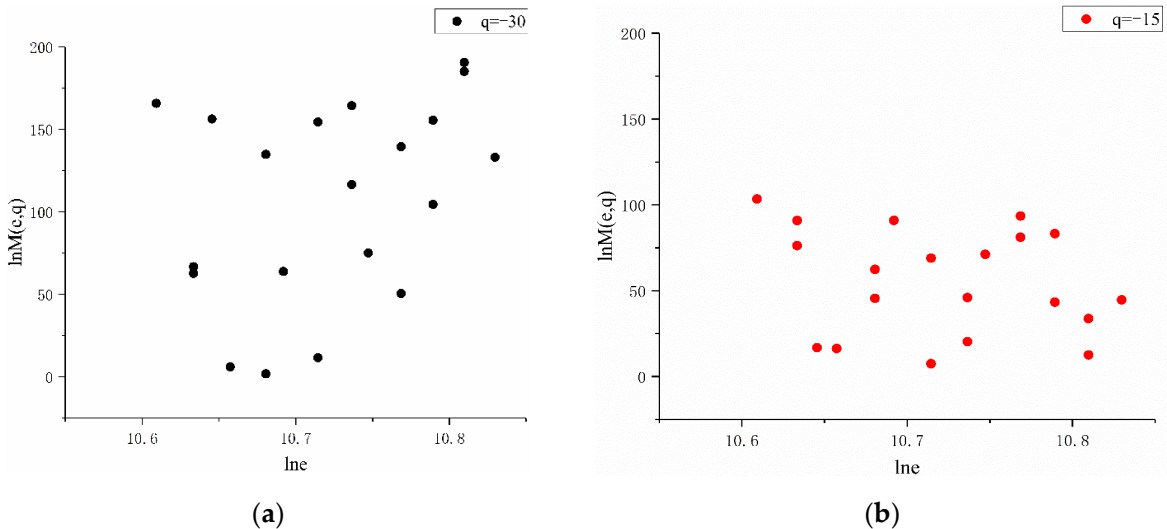

**Figure A3.** The relationship of $\ln M(e, q)$ and $\ln e$ in the Weihe River Basin. (**a**) The relationship of $\ln M(e, q)$ and $\ln e$ when $e$ is [40,000, 50,000] and $q$ is $-30$; (**b**) The relationship of $\ln M(e, q)$ and $\ln e$ when $e$ is [40,000, 50,000] and $q$ is $-15$.

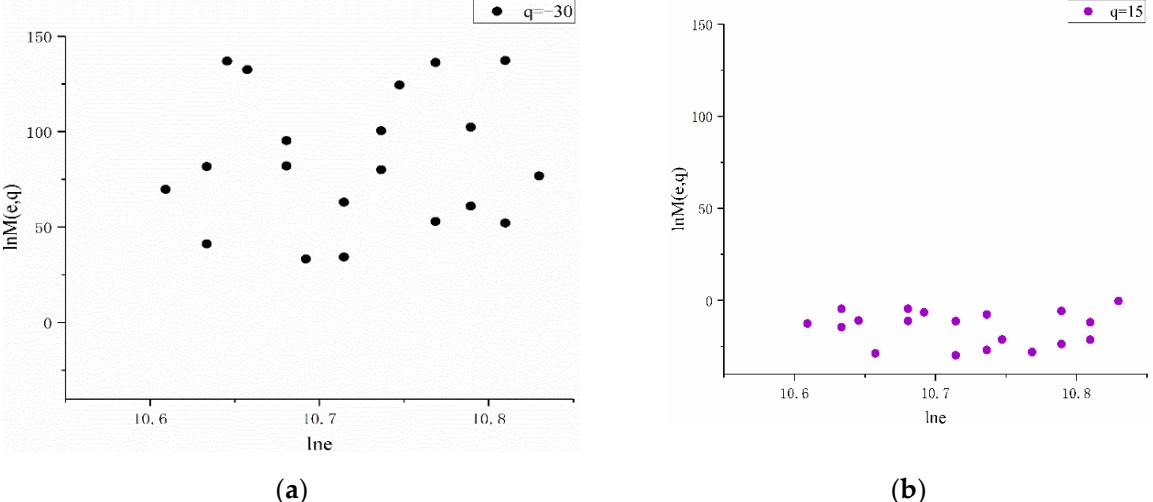

**Figure A4.** The relationship of $\ln M(e, q)$ and $\ln e$ in the Yellow River Downstream Basin. (**a**) The relationship of $\ln M(e, q)$ and $\ln e$ when $e$ is [40,000, 50,000] and $q$ is $-30$; (**b**) The relationship of $\ln M(e, q)$ and $\ln e$ when $e$ is [40,000, 50,000] and $q$ is 15.

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
