# Peer review of "Multifractal Characteristics Analysis Based on Slope Distribution Probability in the Yellow River Basin, China"

_ijgi, doi:10.3390/ijgi10050337_

Round 1
Reviewer 1 Report
Dear Editor, apologize for my delay.
I have finished my review on the proposed paper “Multifractal Characteristics Analysis of Geomorphometry Based on Slope Distribution Probability in the Yellow River Basin, China”, ijgi-1171894-.
Regarding my scientific field I proposed these reviews:
Summary of the manuscript:
The proposed paper deals with the computation and analysis of the slope distribution probability in the Yellow River Basin with a multifractal technique. The authors’ goals were to test the feasibility of multifractal theory in large-scale basin topography, and to analyze the geomorphometrical features of the Yellow River Basin and its sub-basins. The proposed methodology is effective, providing flexibility.
The study provides General review:
The proposed paper is very well written with very good use of English language. This paper is written with a very qualitative scientific style. The proposed paper is very well structured. It begins with an analytical Introduction with the appropriate references that helps the reader to get into the subject immediately. In Introduction, the authors provide literature from previous researches with similar scientific content. Authors describe and set very well the scientific problem and how other researchers have approached it. At the end of the Introduction, authors clearly state the goals of the research.
The methodology is well explained, there are references for the methods that had been applied in the research and other researches could easily repeat this experiment. The results are very well stated with the presence of tables and figures. In my opinion tables and figures are easily understandable and I think there is no need for changes.
The quality of the work in Discussion is acceptable and Conclusions are appropriate for this paper.
Points for revision:
In my opinion, the proposed paper could be characterized as a high-quality research work, complies with aims of International Journal of Geo-Information.
2.2. Data Description
Page , line 109: ” … resolution of 30 m” is it suitable for this work? Add literature.
In “Discussion” or “Conclusions” you should add a paragraph with the applicability limitations of the proposed methodology. Someone expect to see if it possible to apply this methodology in different scale and other basins or watersheds and countries with different climate and geomorphology conditions.
Author Response
Dear Reviewer,
Thank you for your comments concerning our manuscript entitled “Multifractal Characteristics Analysis of Geomorphometry Based on Slope Distribution Probability in the Yellow River Basin, China” (ID: ijgi-1171894). We appreciate the time and effort that you dedicated to providing feedback on our manuscript and are grateful for the insightful comments. We have studied comments carefully and have made the correction which we hope meet with approval.
The specific corrections are as follows:
- line 124:” … resolution of 30 m”, it is suitable for this work of analyzing the geomorphometrical features, and the relevant literatures have been added.
- Considering the Reviewer’s suggestion, the section about the applicability limitations of the proposed methodology has been added in “4. Discussion”. This subsection can be found in “4.3 Limitations and Outlook” of the manuscript.
Once again, thank you very much for your warm work.
Best regards.
Reviewer 2 Report
This manuscript presents an interesting application of multifractal analysis to characterize the texture and landforms of the Yellow River Basins. The explanation of the methodology is very thorough, and the conclusions are concordant with the processing results.
The figures have a good quality and they are all necessary. Only Figure 8 needs the labels a, b and c .
There is the recommendation to the authors that in the discussion they include the explanation how the multifractal analysis improves the results that would have been obtained from a general geomorphometry technique, and how results from the application of both methods compare in quality and difficulty of application.
Author Response
Dear Reviewer,
Thank you for your comments concerning our manuscript entitled “Multifractal Characteristics Analysis of Geomorphometry Based on Slope Distribution Probability in the Yellow River Basin, China” (ID: ijgi-1171894). We appreciate the time and effort and are grateful for the insightful comments. Those comments are valuable and very helpful for revising and improving our paper. We have studied comments carefully and have made the correction which we hope meet with approval. The main corrections are as follows:
- Figure 8 has been labeled a, b, and c. We are sorry for this negligence.
- The explanations of how the multifractal analysis improves the results, and how results from the application of both methods compare in quality and difficulty were added in the section of discussion. Specific corrections can be found in “4.2 Common Topographic Parameters and Extra Multifractal Parameters” of the manuscript.
Once again, thank you very much for your warm work.
Best regards.
Reviewer 3 Report
General comments to the manuscript:
The authors propose in their work the application of a methodology that began to be applied at the end of the 80s, the multifractal analysis, for the study of a morphometric variable (slope), in a large hydrographic basin in China. Since then, this methodology has been applied in numerous works in the field of geomorphology and in particular to the study of basin morphometry. This research paper brings as novelty value its application to a large basin, of almost 800,000 km2, which presents a great diversity in relief. The work is quite well written and very well illustrated and presents a correct structure, although in the opinion of this reviewer, the main drawbacks are the following:
1) In such a large and diverse basin, the proposed methodology would have yielded better results if a greater number of sub-basins had been considered, which would have provided a greater and more solid statistical base. In addition, the authors have not justified nor stated the criteria (physiographic, geological, administrative, etc) applied to establish the watershed division considered in their work. A sufficient number of sub-basins would have allowed obtaining a spatial distribution of the parameters calculated and a more rigorous comparison with respect to traditional geomorphological parameters. This would have led to more robust and interesting conclusions.
2) In Materials and Methods section, the authors should have included a subsection related to traditional geomorphological parameters, their definition and the calculation procedures. Actually, a work stage not mentioned in this section is the comparative study of such parameters with those obtained from the multifractal analysis. The authors include part of these aspects in section 4.2, but they should be included in section 2. Another important issue is that in section 2.2, the original data source, its anomalies and previous correction treatments (if applied) has not been adequately stated, although the site from which they were downloaded is indicated.
3) The way in which the authors explain the multifractal analysis methodology is tedious and the narrative difficult to follow for a non-specialized reader in this field. This constitutes an obstacle that needs to be overcome and that is not entirely attributable to the authors. In spite of this, they should make an effort to convey their work in a simpler and more rigorous way, as well as the fundamentals of the methodology, including definitions, examples and relating the concepts.
4) This reviewer considers that there is a lack of connection between the results obtained from the application of the proposed methodology and their interpretation and physical meaning. The results from multifractal analysis must be contrasted with those obtained from conventional methodologies. In this regard, the comparative study considering the traditional geomorphological parameters is insufficient. The authors should make an effort to convey the advantages and, where appropriate, the disadvantages of their approach. What contribution or step forward does the methodology represent? This is not made explicit in the text, apart from the generalities that the authors mention such as: "the relief of the studied basin is complex, and multifractal analysis provides an effective method to quantitatively express their characteristics". After reading the manuscript, it is still difficult to extract which are the advantages or benefits of this methodology, compared to other traditional techniques that alone or in combination, can provide robust and widely supported by numerous experiences. The authors should make a greater effort in this regard.
5) Similarly, the manuscript would benefit from some final reflections on the applicability of their approach to other basins and some insights on gaps or future improvements that are needed/expected in the field of fractal characterization of river basins.
6) Please see the annotations included in the manuscript.

Author Response
Dear Reviewer,
Thank you for your comments concerning our manuscript entitled “Multifractal Characteristics Analysis of Geomorphometry Based on Slope Distribution Probability in the Yellow River Basin, China” (ID: ijgi-1171894). We appreciate the time and effort and are grateful for the insightful comments. Those comments are valuable and very helpful for revising and improving our paper. We have studied comments carefully and have made the correction which we hope meet with approval. The main corrections are as follows:
- In order to increase the credibility and statistical significance of this analysis, the three-level sub-basins of the Yellow River Basin (a total of 31) were selected for analysing, and we calculated the topographic parameters and multifractal parameters of these regions respectively, and analyzed the relationship between them. The criteria applied to establish the watershed division was provided by the National Cryosphere Desert Data Center, and these sub-basins were divided according to the catchment area.
- In Materials and Methods section, the subsection about the definition and the calculation procedures of geomorphological parameters was added in “2.3. Multifractal Approach and Traditional Geomorphological Parameters”. In addition, the original data source was indicated, and the data has been corrected for this study, which has been indicated in this paper.
- In order to explain the results of multifractal analysis more clearly, a summary table was added, which include the location for each basin, and the description of the geomorphological implications in all of the sub-basins. We hope that this will make the results better understood and accepted.
- In order to better explain the applicability of multifractal theory in this paper, the subsection of “4.3. Limitations and Outlook” in the section of discussion was added, which explained the limitations of method of the research, and how it will be improved in future work.
- The specific corrections can be found in the attachment.
Once again, thank you very much for your warm work.
Best regards.

Round 2
Reviewer 3 Report
After carefully reading the manuscript, I must acknowledge the effort made by the authors to improve the work. They have added new sections and explanatory tables, which contribute to a better understanding of the overall paper and have improved the consistency throughout the text by homogenizing the use of acronyms, definitions and other terms. Nevertheless, there are some aspects that the authors have overlooked and still need improvement:
- Subbasin division criteria: The authors mention the organization who established this basin division, which is interesting, but they still do not mention the specific criteria applied for carrying out said division. This is an important aspect that should be included.
- I am not quite sure why the authors have removed the sentence regarding the ASTER GDEM. One of the premises of any scientific work is that it must be replicable, and for this, readers need to know the data source used. I suggest keeping the indication of the data source (origin). On the other hand, the authors affirm that this data have been corrected, but they do not specify how. Which was the correction process applied? How does the processed product improve compared to the raw one?
- Lines 100-111: I appreciate the fact that the authors have included information on the geological context of the study area, however, the narrative should be improved. In its current state, it looks like a succession of long sentences enumerating names and adjectives, and the overall meaning is difficult to extract.
- Table 2. Very nice and helpful from a reader point of view.
- The authors have added section 2.3 “Multifractal approach and geomorphological parameters”, what I´m sure is going to be very appreciated by potential readers, especially by those with different backgrounds. Nevertheless, the authors begin directly by talking about multifractals, but ….wouldn't it be more appropriate to start by defining what a monofractal is, and then continue talking about multifractals? The narrative line of this paragraph needs to be improved. What about the singularity exponent?
- Section 4.3. Limitations and outlooks. I suggest to relocate this section and placing it within section 5. Besides, there are a few English corrections to be made:
Line 503: replace “the method of this paper” by “the method proposed”
Line 504: replace “of any scale basin……” by “basins of any scale”
Line 504-505: replace “due to the data that are available are limited” by “since data availability is limited”
Line 505: replace “was used for calculating in this paper” by “was used to perform calculations, however, the study would have benefitted from the use of data with higher resolution”
Line 508-511: I suggest to replace the line “The results of multifractal analysis are meaningful only when they are calculated in the free-scale range. It is necessary to….” by “The results of multifractal analysis are meaningful only when they are calculated in the free-scale range, being therefore necessary its previous calculation”.
Abstract: At the end of the abstract, the authors need to back their affirmation “ there is a strong correlation between common topographic parameters and multifractal parameters” with numbers. I suggest indicating some of the correlation values obtained.
Other corrections:
- Line 46: enables to
- Line 276: Overhanging river definition, my suggestion: As the Yellow River Downstream flows gently, the “overhanging river” (fluvial sections where the river bottom has risen above the ground level as result of silting) and deltas were formed.
Author Response
Dear Reviewer,
Thank you for your comments concerning our manuscript entitled “Multifractal Characteristics Analysis of Geomorphometry Based on Slope Distribution Probability in the Yellow River Basin, China” (ID: ijgi-1171894), those comments are very helpful for improving our paper. We appreciate your time and effort and are sorry for our negligence. We have studied comments carefully and have made the correction which we hope meet with approval. The specific corrections can be found in the attachment.
The main corrections are as follows:
- The sub-basin division criteria were carried out based on the catchment area of the basins, and sub-basins studied are the secondary basins with the largest catchment area in the Yellow River Basin. These criteria have been included in the paper.
- The data source and the description of data processing were added in the paper. The data was processed by raster mosaic and outlier correction, which is suitable for the study of this paper. These data can be downloaded according to the website link given in the paper, which make this work replicable.
- We are sorry for our narrative about geological context of the study area. In order to better describe the geological context of the Yellow River Basin, this part has been corrected.
- Considering the reviewer’s comments, the section of “2.3 Multifractal approach and geomorphological parameters” has been corrected. At the beginning of this section, the mono-fractal was first introduced, and then the multifractal was explained. The explanation of the singularity exponent was added in this section.
- This section of “4.3. Limitations and outlooks” has been relocated, and it can be found in section “5. conclusion”. According to reviewer’s suggestions, some English corrections has been made.
- The correlation values between some common topographic parameters and multifractal parameters were added in the end of the abstract.
Once again, thank you very much for your warm work.
Best regards.

This manuscript is a resubmission of an earlier submission. The following is a list of the peer review reports and author responses from that submission.
Round 1
Reviewer 1 Report
Dear Authors,
Your manuscript needs some modifications in several parts. There are parts and phrases that are repeated. English needs your attention. Specific suggestions can be found highlighted in the attached document.

Author Response
Dear Reviewer,
Thank you for your comments concerning our manuscript. We appreciate the time and effort that you dedicated to providing feedback on our manuscript and are grateful for the insightful comments on and valuable improvements to our paper. We have studied comments carefully and have made the correction which we hope meet with approval. The revised portion is highlighted in the attachment.
Please see the attachment.
Once again, thank you very much for your warm work.
Best regards.

Reviewer 2 Report
To start with, I would like to say that this is a well written paper, some minor comments can be seen bellow.
The problem that I am having with that paper is that I cannot see the actual point that the authors are trying to make. Yes the river is large and yes the fractal theory allows to analyses it, and it in general matches the geomorphology. But why was the study done? For automatisation of observation of such a landforms. And if so how does it improve on other teledetection methods. Is it the same accuracy as filed work observation or other methods – we do not know that. We know that is based on 30m DEM but that does not give us an inside to the accuracy of this method of detecting geomorphology. Some information on that is given in 4.1 but I do not think it is enough. This is why I am recommending a revision.
L13” researches were” this does not need to be in plural
L25 scientific basis for analyzing”
L32-33 “
Although the morphology 32 of landforms is complex and changeable, it can be found by using fractal analysis that 33 they are still subject to mathematical laws.” This is an oversimplification, needs to be phrased better. They are not subjected to mathematical laws, forming processed can be explained or described by a special type of mathematical equations laws and theorems.
L39 Later to what?
L41 compared to
L54 What is “multifractal technology” ?
L102 descriptions should be bigger
L111 space after 10.4
L242 forming deltas
Author Response

(The authors gave the same response as above.)

Reviewer 3 Report
Dear Authors,
All my comments are in attachments.
As a geomorphologist, I referred only to the part concerning the relief. I am not able to check the mathematical elements in terms of merit.
Geomorphology is a science about Earth landforms. You present relief, surface and landforms of the Yellow River Basin. In all text geomorphology, landforms, terrain, topography and surface are accidentally mixed. This creates factual mistakes.
Main comments:
Tittle is inaccurate. It should contain geomorphometry, because quantitative characteristics in geomorphology is geomorphometry, especially if you use mathematical equipments! This notion should be exist in the text!
Geomorphological terms are notoriously misused in the text (eg. which is also the difference distribution of geomorphology). Geomorphology is science about Earth landforms. You present relief, surface and landforms of Yellow River Basin. Geomorphologic features are:
morphology, morphometry, morphogenesis and morphochronology. Two last elements are not exist in the text. You describe only morphometry! If you used DEM is part of geomorphological features of landforms is named as GEOMORPHOMETRY. I gave a link in the text...
Next mistake: what is topographic relief (tabel 2)? Maybe it is the usual altitude/height above sea level? It is not clear.
Detailed comments:
No information about the height of the source. Neither about the maximum height of the river level in the study area. This gives an idea of the height difference.
If you compare the relationships in figure 4 you must unify the 0Y axis from value -600 to 600 in all diagrams!
If you compare the relationships in figure 6 you must unify the 0Y axis from 1.2 to 3.0 value in a and b diagrams!
If you compare the relationships in figure 8 you must unify the 0Y axis from 100 to 400 value in b and c diagrams!
In Figure 9 please unify both axes.
I hope that my comment will be substantial...
Regards.

Author Response
Dear Reviewer,
Thank you for your comments concerning our manuscript. We appreciate the time and effort that you dedicated to providing feedback on our manuscript and are grateful for the insightful comments on and valuable improvements to our paper. Those comments are all valuable and very helpful for revising and improving our paper. We have studied comments carefully and have made the correction which we hope meet with approval. The revised portion is highlighted in the attachment.
Please see the attachment.
Once again, thank you very much for your warm work.
Best regards.

Round 2
Reviewer 1 Report
Dear Authors,
Please find some minor suggestions included in the document attached. Please pay attention to the English syntax and grammar. Several parts need your attention especially after the modifications from the previous version. I feel most of the time dedicated in the manuscript should be focused on that. I would propose next time to provide also the manuscript with accepted changes in order to be easily readable for the reviewers.

Author Response
Dear Reviewer,
Thank you for your comments concerning our manuscript entitled “Multifractal characteristics analysis of geomorphometry based on slope distribution probability in the Yellow River Basin, China” (ID: ijgi-1103309). Your comments are valuable and very helpful for revising and improving our paper, as well as the important guiding significance to our research. Considering the Reviewer’s suggestions, our manuscript has been checked by our colleagues who are native English speakers. We have studied comments carefully and have made the correction which we hope meet with approval. The revised portion is highlighted in the paper. The specific corrections can be found in the attached document.
Once again, special thanks to you for your warm work.
Best regards.

Reviewer 2 Report
Since the authors explained my doubts I do believe that the manuscript can now be published.
Author Response
Dear Reviewer,
Thanks very much for your kind work and consideration on our manuscript. We deeply appreciate your recognition of our research work.
Once again, thank you for your warm work.
Best regards.